# Hierarchical Trajectory Planning for Narrow-Space Automated Parking with Deep Reinforcement Learning: A Federated Learning Scheme

**DOI:** 10.3390/s23084087

**Published:** 2023-04-18

**Authors:** Zheng Yuan, Zhe Wang, Xinhang Li, Lei Li, Lin Zhang

**Affiliations:** 1School of Artificial Intelligence, Beijing University of Posts and Telecommunications, 10 Xitucheng Road, Haidian Distinct, Beijing 100876, China; yuanzheng@bupt.edu.cn (Z.Y.); lixinhang@bupt.edu.cn (X.L.); leili@bupt.edu.cn (L.L.); 2Centre for Telecommunications Research, King’s College London, London WC2R 2LS, UK; tylor.wang@kcl.ac.uk

**Keywords:** automated parking, trajectory planning, federated deep reinforcement learning, nonlinear optimization

## Abstract

Collision-free trajectory planning in narrow spaces has become one of the most challenging tasks in automated parking scenarios. Previous optimization-based approaches can generate accurate parking trajectories, but these methods cannot compute feasible solutions with extremely complex constraints in a limited time. Recent research uses neural-network-based approaches that can generate time-optimized parking trajectories in linear time. However, the generalization of these neural network models in different parking scenarios has not been considered thoroughly and the risk of privacy compromise exists in the case of centralized training. To address the above issues, this paper proposes a **h**ierarchical tr**a**jectory p**l**anning method with deep reinf**o**rcement learning in the f**e**derated learning **s**cheme (HALOES) to rapidly and accurately generate collision-free automated parking trajectories in multiple narrow spaces. HALOES is a federated learning based hierarchical trajectory planning method to fully exert high-level deep reinforcement learning and the low-level optimization-based approach. HALOES further fuse the deep reinforcement learning model parameters to improve the generalization capabilities with a decentralized training scheme. The federated learning scheme in HALOES aims to protect the privacy of the vehicle’s data during model parameter aggregation. Simulation results show that the proposed method can achieve efficient automatic parking in multiple narrow spaces, improve planning time from 12.15% to 66.02% compared to other state-of-the-art methods (e.g., hybrid A*, OBCA) and maintain the same level of trajectory accuracy while having great model generalization.

## 1. Introduction

Automated parking is a hot issue of interest in academia and industry. Autonomous vehicles (AVs) can improve driving safety, efficiency and convenience through Advanced Driving Assistance Systems (ADAS) [1]. Automated parking is an essential application of ADAS for autonomous vehicles, which has been used by many car manufacturers such as Audi, BMW, Mercedes-Benz and BYD [2]. In recent years, parking space has become scarce in many cities with the increase in vehicles and the relative lag in infrastructure. The narrow and crowded parking environment increases the risk of collision and makes parking more difficult for drivers while also bringing new challenges to automated parking technology [3]. This paper aims to achieve a more efficient collision-free trajectory planning scheme for automated parking in narrow parking spaces (Figure 1).

Trajectory planning is a critical component of autonomous vehicles, enabling the generation of a collision-free, smooth and kinematically feasible curve for the vehicle’s motion. Compared to the trajectory planning of on-road autonomous driving, the trajectory planning of automated parking has more challenges. To be specific: (1) Polynomial-based path planning approaches are not suitable for automated parking because parked path curves can have non-guidable points, whereas polynomial-based methods can only generate smooth, reachable path curves. (2) Obstacle avoidance constraints in automated parking are more complex than those encountered in on-road autonomous driving due to the intricate parking environment. (3) Automated parking requires full use of the vehicle’s steering capability compared to the relatively smooth curve of on-road autonomous driving [4]. Currently, research mainly has three types of trajectory planning for automated parking: the sample-and-search-based method, the optimization-based method and the neural-network-based method.

Sample-and-search-based path planning discretizes the continuous state space into a graph with nodes and searches the graph structure for the optimal path linking the start and goal points. The typical methods of node-based search with sampling methods are RRT and RRT*, which can relatively quickly generate a curve to the goal point. However, these methods have no guarantee of kinetic feasibility, resulting in no reliable trajectories [5]. D. Dolgov et al. proposed the Hybrid A* algorithm, which discretizes and samples the control space to obtain a smooth and kinetic feasible trajectory curve [6]. He et al. proposed a fast A* anchor point-based path planning algorithm based on Hybrid A* for solving reverse parking path planning in narrow spaces [7]. The sample-and-search-based method can be applied to a wide range of scenarios. However, balancing computational accuracy and time is a challenge because low sampling resolution can lead to inaccurate path planning, while high sampling resolution can consume too many computational resources.

The optimization-based trajectory planning algorithms such as Model Predictive Control (MPC) have been applied to unmanned aircraft and autonomous vehicles. In general, the trajectory planning obstacle avoidance problem is NP-hard and the optimization-based methods consider the obstacle avoidance problem as an Optimal Control Problem (OCP), which is solved using the technique of Nonlinear Programming (NP) [8]. This category has two branches, that is, soft constraint-based and hard constraint-based optimizations. On soft constraint-based methods, K Shibata et al. designed an Artificial Potential Field (APF) in the cost function as the soft constraint by considering the obstacle as a mass but cannot guarantee collision avoidance [4]. On hard constraint-based methods, Zhang et al. proposed an Optimization-Based Collision Avoidance (OBCA) algorithm by reformulating a smooth obstacle avoidance constraint containing the vehicle geometry [8]. Li et al. proposed a unified OCP model aimed at unstructured parking spaces using a triangle-area criterion to construct the collision-avoidance constraints [9]. However, when parking space is narrow, the feasible domain will be greatly limited due to collision avoidance constraints, resulting in non-linear programming solvers that cannot compute a feasible solution in a finite time.

The neural-network-based method is proposed to generate parking trajectories in narrow scenarios while keeping the time-optimality of the trajectory and low computational overhead. Few studies use neural-network-based methods in automated parking trajectory planning [10,11,12,13,14]. Zhang et al. [15] used Deep Deterministic Policy Gradient (DDPG), a deep reinforcement learning method, to solve the perpendicular parking problem. However, this work focuses only on perpendicular parking and does not consider speed planning. Song et al. [16] proposed a method to solve the parallel parking problem by fusing Nonlinear Programming (NP) and Monte Carlo Tree Search (MCTS), using NP to generate data and train the policy neural network offline, followed by using the policy network to guide MCTS to complete the trajectory planning. The above neural-network-based method can trade off computational accuracy and computational time. However, these methods need to be more generalized and applied to other scenarios because it is only trained in a single scene.

In the field of Intelligent Transportation Systems (ITSs), Federated Learning (FL) has emerged as a promising approach for collaborative model training without the need for lengthy data transfers or sacrificing user privacy. By leveraging the distributed computing power of participating devices, federated learning enables the collaborative training of a globally shared vehicle AI model, which can be utilized by all participating devices for improved performance and efficiency. In recent years, federated learning has gained significant attention from the research community due to its potential to enhance the effectiveness and privacy of ITS [17,18]. For privacy protection in connected vehicles, federated learning has been applied to several intelligent transportation systems to achieve traffic flow prediction, parking reservation and traffic edge computing without privacy leakage [19,20,21,22]. Moreover, the application of Federated Reinforcement Learning (FRL) has extended beyond ITSs to encompass a variety of industrial Internet of Things (IoT) applications, including robot task scheduling [23] and robot navigation tasks [24]. FRL has demonstrated improved generalization and faster convergence of neural network models, enabling the efficient and effective training of complex reinforcement learning models across distributed devices.

In this paper, we propose a hierarchical trajectory planning method with deep reinforcement learning in the federated learning scheme, which is named HALOES, to rapidly and accurately generate collision-free automated parking trajectories in multiple narrow spaces. The FL method considers the computation time and the vehicle kinematic constraints for automated parking trajectory planning in extremely narrow spaces. The contributions of this paper are as follows:A novel hierarchical trajectory planning method with deep reinforcement learning and optimization-based approach integration is proposed to achieve computational accuracy and computational time trade-off. The method has a high-level neural network model for rapid reference trajectory generation and a low-level optimization model to refine the trajectory.A decentralized training scheme is introduced in the model training module to improve the generalization of model performance by fusing the model parameters trained in different parking scenarios and the federated learning scheme is used in decentralized deep reinforcement learning to protect the privacy of the vehicle’s data during model parameter fusion.Simulation results demonstrate that the proposed HALOES method can enable efficient automated parking in narrow spaces and outperforms other state-of-art methods, such as Hybrid A* and OBCA, in terms of planning time, trajectory accuracy and model generalization.

The rest of the paper is as follows. Section 2 presents the kinematics of vehicles, the formulation of model predictive trajectory planning and the background of deep reinforcement learning. The details of HALOES are shown in Section 3. In Section 4, the federated learning scheme is presented. Simulations on several narrow-space parking and detailed analyses of the simulation results are presented in Section 5. Finally, conclusions are drawn in Section 6.

## 2. Materials and Methods

This section will introduce the backgrounds of deep reinforcement learning and federated learning.

### 2.1. Deep Reinforcement Learning

Instead of sampling-based and optimization-based approaches, Deep Reinforcement Learning (DRL) seeks to find the optimal policy function $\pi^{*}$ with neural network parameter θ [25]. The DRL approach starkly contrasts the model predictive trajectory planning method, as it can achieve model-free operation. In DRL, the agent operates without relying on a predefined environment model. Instead, the agent of DRL interacts directly with the environment by making decisions based on the observation of the current state and the policy function. The basic DRL problem is modeled as a Markov Decision Process (MDP), which can be defined by a tuple of elements <S,A,P,r,H,γ>, in which S is the set of states, A is the set of actions, P:S×A→S maps an action and the current state to the next state, r:S×A→R is the reward function that maps an action and a state to a scalar, *H* is the horizon and γ∈(0,1] is a discount factor of reward. At the timestep *t*, the agent of DRL uses the policy function to determine which action $a_{t}\in A$ to take. The policy function maps the current state to a probability distribution over the action space. Then the environment updates to the next state based on the transition function $s_{t+1}=P\left(s_{t},a_{t}\right)\in S$ and returns a scalar reward $r_{t}=r\left(s_{t},a_{t}\right)$ to the agent for its action. The discounted reward from the state $s_{t}$ is $R_{t}\left(s_{t},a_{t}\right)=\sum_{i=0}^{H}\gamma^{i}r\left(s_{t+i},a_{t+i}\right)$. The ultimate goal of DRL is to find the optimal policy function that maximizes the expected discounted return $\underset{\theta }{\mathrm{argmax}}E\left[R_{1}\left(s_{t},a_{t}\right)|S_{t}∼\rho^{\pi_{\theta }},a_{t}∼\pi_{\theta }\right]$, in which $\rho^{\pi_{\theta }}$ is the state visitation distribution under the policy function $\pi_{\theta }$.

### 2.2. Federated Learning

FL is a promising and innovative approach to distributed collaborative Artificial Intelligence (AI) that differs significantly from traditional methods [26]. Unlike centralized approaches, which often require vast amounts of sensitive data to be centralized and shared among multiple participants, FL allows for collaborative training across multiple participants through a centralized server while preserving the privacy and security of the individual actual data. In doing so, FL provides a promising solution for distributed Deep Reinforcement Learning (DRL) that can protect the security and privacy of the distributed DRL participants.

The general FL is divided into two distinct stages, each of which plays a critical role in the training process. The first stage is the distributed local training and update; before starting the training, the server initializes a new neural network model, such as $W_{g}^{0}$, and transmits it to the other participants to start the distributed training; each participant $w_{m}$ trains a local neural network model using its own dataset and update $w_{m}$ by minimizing the loss function $\mathrm{argmin}F(w_{m}),m\in M$, in which *M* is the set of distributed DRL participants and $F(w_{m})$ is different in different FL algorithms.

The second stage is model aggregation and downloading. The server aggregates all models into a new version of the global model after updated model collection from all the FL participants; the aggregate process could be defined as $W_{g}=\left(1/|M|\right)\sum w_{m}$, after all the updated models collected from the participants; note that the model aggregation method differs for different FL algorithms. The participants download the new global model for optimizing the local model in the next learning round.

## 3. Hierarchical Trajectory Planning with Deep Reinforcement Learning

This section will introduce the kinematic vehicle models used by HALOES and the overall scheme of HALOES.

### 3.1. Kinematic Vehicle Model

This paper employs the Kinematic Single-track (KS) model as the kinematic vehicle model. As shown in Figure 2, in the x−y coordinate system, the reference point of the vehicle is the rear-wheel axle mid-point ${\left[S_{x},S_{y}\right]}^{T}$, and the orientation in the global coordinate system is Ψ. *v* represents the current velocity, δ represents the steering angle, $l_{WB}$ is the wheelbase of vehicle, $l_{R}$ is the vehicle rear hang length, $l_{F}$ is the vehicle front hang length and $l_{W}$ is the vehicle width. According to the KS model, the state transfer equation of the vehicle is as follows,
(1)$$\left[\begin{array}{c} S_{x}\left(t+\Delta t\right)\\ S_{y}\left(t+\Delta t\right)\\ v(t+\Delta t)\\ \Psi (t+\Delta t)\\ \delta (t+\Delta t) \end{array}\right]=\left[\begin{array}{c} S_{x}\left(t\right)\\ S_{y}\left(t\right)\\ v(t)\\ \Psi (t)\\ \delta (t) \end{array}\right]+\left[\begin{array}{c} v(t)\cos\Psi (t)\\ v(t)\sin\Psi (t)\\ a(t)\\ \frac{v(t)}{l_{WB}}\tan\delta \left(t\right)\\ v_{\delta }\left(t\right) \end{array}\right]\Delta t,$$
in which *t* and Δt represent the current time and the control interval, respectively. a(t) and $v_{\delta }\left(t\right)$ are the control profiles, representing the acceleration and angular velocity of steering angles at the current time, respectively.

Traditional optimization-based approaches typically use model predictive trajectory planning. Model predictive trajectory planning can be considered as an OCP, which can be solved via an NP solver. Let $x\in R^{n_{x}}$ be the vehicle state profile and $u\in R^{n_{u}}$ the control profile; the details of x and u at time *t* are as follows:(2)$$x\left(t\right)=[S_{x}\left(t\right),S_{y}\left(t\right),v\left(t\right),\Psi \left(t\right),\delta \left(t\right)].$$
(3)$$u\left(t\right)=[a\left(t\right),v_{\delta }\left(t\right)].$$

Let k∈{1,…,N} represent the discrete time index with time interval Δt; the state transfer Equation (Equation 1) can be written as x(k+1)=x(k)+f(x(k),u(k))·Δt, in which f(x(k),u(k)) denotes the differential equation of vehicle state x specified which could be written as follows,
(4)$$f\left(x\left(k\right),u\left(k\right)\right)=\frac{d}{dt}\left[\begin{array}{c} S_{x}\left(k\right)\\ S_{y}\left(k\right)\\ v(k)\\ \Psi (k)\\ \delta (k) \end{array}\right]=\left[\begin{array}{c} v(k)\cos\Psi (k)\\ v(k)\sin\Psi (k)\\ a(k)\\ \frac{v(k)}{l_{WB}}\tan\delta \left(k\right)\\ v_{\delta }\left(k\right) \end{array}\right]$$

The finite-horizon OCP with collision avoidance constraint can be written as:
(5a)$$\begin{array}{cc} \; & \min_{x,u}\sum_{k=1}^{N}J\left(x\left(k\right),u\left(k\right)\right), \end{array}$$
(5b)$$\begin{array}{cc} s.t.\;\; & \mathrm{kinematic}\;\mathrm{principles}\;(\mathrm{Equation}\;\;(1)) \end{array}$$
(5c)$$\begin{array}{cc} \; & \mathrm{collision}\text{-}\mathrm{free}\;\mathrm{conditions} \end{array}$$
(5d)$$\begin{array}{cc} \; & \mathrm{for}\;\mathrm{all}\;k\in \{1,\ldots ,N\}. \end{array}$$
in which *N* represents the prediction horizon, and *J* is the cost function to penalize deviations from the target point. Equation (5d) is the obstacle avoidance constraint to achieve the obstacle avoidance trajectory.

### 3.2. DRL-Based Trajectory Planning for Automated Parking

Automated parking trajectory planning can be modeled as a Markov process. This section describes the formulation of the DRL for automated parking trajectory planning and the training process of the DRL model.

#### 3.2.1. Setup of the DRL-Based Trajectory Planning

In this section, we define the setup of the DRL framework, such as the set of state S, the set of action A and the reward function *r*.

Automated parking trajectory planning in a narrow space requires constant consideration of vehicle coordinates, target point coordinates, obstacle location information and vehicle kinematic states such as speed and steering wheel angle. In this paper, we define the observed state at the moment of time step *t* as:(6)$$S^{t}=\left\{P_{e}^{t},{\dot{P}}_{g}^{t},v_{e}^{t},\delta_{e}^{t},V_{obs}^{t}\right\},$$
in which $P_{e}^{t}=\left\{S_{x,e}\left(t\right),S_{y,e}\left(t\right),\Psi_{e}\left(t\right)\right\}$ presents the heading coordinates of the ego vehicle at the time step *t*, ${\dot{P}}_{g}^{t}=P_{g}^{t}-P_{e}^{t}=\left\{S_{x,g}\left(t\right)-S_{x,e}\left(t\right),S_{y,g}\left(t\right)-S_{y,e}\left(t\right),\Psi_{g}\left(t\right)-\Psi_{e}\left(t\right)\right\}$ denotes the position of the target point relative to the ego vehicle, $S_{x,g}\left(t\right),S_{y,g}\left(t\right)$ and $\Psi_{g}\left(t\right)$ represent the coordinates of the two-dimensional vehicle target point (x,y) and the heading angle, respectively. $v_{e}^{t}$ and $\delta_{e}^{t}$ denote the speed and steering wheel angle of the ego vehicle at time step *t*, respectively. $V_{obs}^{t}=V_{obs1}^{t},V_{obs2}^{t},\ldots ,V_{obsN}^{t}$ is the set of relative positions of the obstacles and the ego vehicle, in which $V_{obsN}^{t}=\left\{v_{obs,x}^{1}-S_{x,e}\left(t\right),v_{obs,y}^{1}-S_{y,e}\left(t\right),\ldots ,v_{obs,x}^{|V_{obsN}\;\;|}-S_{x,e}\left(t\right),v_{obs,y}^{|V_{obsN}\;\;|}-S_{y,e}\left(t\right)\right\}$ and $|V_{obsN}|$ is the number of vertices of the *N*th obstacle.

Since the vehicle kinetic characteristics, such as maximum steering wheel angle and maximum turning radius, need to be fully utilized in parking, the action output of the neural network needs to conform to the vehicle kinetic constraints. In this paper, in order to conform to the vehicle kinetic characteristics, instead of using the relative change of distance as the vehicle action set, e.g., A={Δx,Δy,ΔΨ}, the acceleration, as well as the steering wheel cornering rate, are used as the action and the action space is defined as:(7)$$A=\left\{\dot{a},\dot{v_{\delta }}\right\},\dot{a}\in \left[-1,1\right],\dot{v_{\delta }}\in \left[-1,1\right].$$
Normalization of the action space improves the learning efficiency of the neural network. The control inputs to the vehicle are $A_{input}=\left\{\dot{a}*a_{max},\dot{v_{\delta }}*v_{\delta ,max}\right\}$ where $a_{max}$ and $v_{\delta ,max}$ are the maximum values of acceleration and steering wheel rate, respectively.

The reward function in deep reinforcement learning mixes multiple reward values to guide the convergence direction of the model. In automatic parking trajectory planning, three aspects of reward need to be considered; one aspect is the duration of parking, one is the distance to the target point and the last one is the collision with the obstacle. In addition, we use negative rewards to penalize the agent in order to make it able to converge faster. The whole reward function is:(8)$$r=-c_{t}*R_{t}-c_{d}*R_{d}-c_{\Psi }*R_{\Psi }-c_{o}*R_{o},$$
where $c_{*}$ is the penalty factor for each item. $R_{t}$ is the time-term penalty as a fixed value. $R_{d}$ is the penalty term for distance to the target point, which represents the change in distance of the vehicle from the target point:(9)$$R_{d}=d_{t}-d_{t-1},d_{t}\leftarrow {\left(S_{x}\left(t\right)-S_{x,g}\left(t\right)\right)}^{2}+{\left(S_{y}\left(t\right)-S_{y,g}\left(t\right)\right)}^{2}.$$$R_{\Psi }$ is the penalty term for the orientation angle to the target point, which represents the change in orientation of the vehicle from the target point:(10)$$R_{\Phi }=\Delta \Psi_{t}-\Delta \Psi_{t-1},\Delta \Psi_{t}\leftarrow \left|\Psi_{g}-\Psi_{t}\right|.$$

$R_{o}$ represents the obstacle collision penalty, which applies a fixed penalty when the vehicle collides with an obstacle. There are two prominent cases where a vehicle collides with an obstacle; one is that at least one vehicle with one vertex of the polygon is located inside the obstacle and the other is that at least one obstacle with a vertex of the polygon is located inside the vehicle. At time *t*, if neither of the two cases exists, the automatic parking trajectory can be considered collision-free with the obstacle. Note that each obstacle polygon should be convex. If the polygonal obstacle is non-convex, it must first be divided into several convex polygons. As shown in Figure 3, the set of vertices of the obstacle is $V_{obs}=\left\{O_{1},O_{2},O_{3},O_{4}\right\}$ and the set of vertices of the vehicle is $V_{veh}=\left\{V_{1},V_{2},V_{3},V_{4}\right\}$. Let any point $Q_{obs}=\left(x_{obs},y_{obs}\right)\in V_{obs}$ denote the coordinates of a vertex of the obstacle and any point $Q_{veh}=\left(x_{veh},y_{veh}\right)\in V_{veh}$ denote the coordinates of a vertex of the vehicle. The collision-free condition can be obtained from the triangular area criterion:
(11a)$$\begin{array}{c} S_{\Delta Q_{\mathrm{veh}}\;O_{4}O_{1}}+\sum_{i=1}^{3}S_{\Delta Q_{\mathrm{veh}}\;O_{i}O_{i+1}}>S_{obs}, \end{array}$$
(11b)$$\begin{array}{c} S_{\Delta Q_{\mathrm{obs}}\;V_{4}V_{1}}+\sum_{i=1}^{3}S_{\Delta Q_{\mathrm{obs}}\;V_{i}V_{i+1}}>S_{veh}, \end{array}$$
where $S_{\Delta }$ denotes the triangle area, and $S_{obs}$ and $S_{veh}$ denote the area of the polygonal obstacle and the area of the polygonal vehicle. Here, the area of a triangle and the area of a polygonal obstacle can be calculated using the shoelace theorem; let the set of vertices of a convex polygon be $V_{convex}=\left\{Q_{1},Q_{2},\ldots ,Q_{N_{con}}\right\}$ where $Q_{i}=\left(x_{i},y_{i}\right)$ denotes the coordinate, then:(12)$$\begin{array}{c} S_{Q_{1},Q_{2},\ldots ,Q_{{N\;}_{con}}}=\frac{1}{2}|\left(\sum_{n=1}^{N-1}x_{n}y_{n+1}+x_{N_{con}}y_{1}\right)\\ -(\sum_{n=1}^{N-1}x_{n+1}y_{n}+x_{1}y_{N_{con}})|. \end{array}$$Thus $R_{o}$ is defined as follows:(13)$$R_{o}=\left\{\begin{array}{cc} 0 & \mathrm{Equation}\;(11a)\;\mathrm{is}\;\mathrm{true}\;\mathrm{and}\;\mathrm{Equation}\;(11b)\;\mathrm{is}\;\mathrm{true}\\ 1 & \;\mathrm{others}\; \end{array}\right..$$

#### 3.2.2. Training Process of DRL Model

The training goal of reinforcement learning is to find the optimal policy function $\pi^{*}$, parameterized by θ, which maximizes the expected discount reward of Equation (Equation 14).
(14)$$\underset{\theta }{\mathrm{argmax}}E\left[\sum_{i=0}^{H}r\left(S_{t},a_{t}\right)|S_{t}∼\rho^{\pi_{\theta }},a_{t}∼\pi_{\theta }\right],$$
where *r* is calculated via Equation (Equation 8). The Deep reinforcement learning is mainly divided into the off-policy method and the on-policy method. The off-policy methods, such as Deep Deterministic Policy Gradient (DDPG) and Soft Actor-Critic (SAC), improve sampling efficiency by maintaining the experience pool, so the model can reuse old data for training. The on-policy methods, such as Proximal Policy Optimization (PPO), optimize the same policy as its own decision network, making the model update more stable by directly optimizing the objective function. The trajectory planning process in complex environments often involves many exploration processes, making it a computationally challenging task. Reinforcement learning (RL) methods utilizing off-policy have shown the potential to enhance the model’s performance and stability. By exploiting these existing data, RL methods using off-policy can make more efficient use of available resources and accelerate the training process. As a result, using off-policy in RL methods can improve the overall performance and robustness of the learned policies, making them well suited for real-world applications. DDPG is an Actor-Critic, model-free algorithm based on the deterministic policy gradient used for learning policies in environments with continuous action spaces. DDPG uses two neural networks, an actor and a critic network, to update the policy and value function, respectively. The actor network is updated using the sampled policy gradient:(15)$$\nabla_{\theta^{\mu }}J=\frac{1}{N}\sum_{i}^{N}\nabla_{a}Q\left(s,a\;|\theta^{Q}\right){{|}_{s=s_{i},a=\mu \left(s_{i}\right)}\nabla_{\theta^{\mu }}\mu \left(s|\theta^{\mu }\right)|}_{s_{i}},$$
where *N* is the mini-batch size, $Q\left(s,a\;|\theta^{Q}\right)$, and $\mu \left(s|\theta^{\mu }\right)$ is the critic network and actor network with parameters $\theta^{Q}$ and $\theta^{\mu }$. The critic network is updated by minimizing the loss:(16)$$L=\frac{1}{N}\sum_{i}^{N}{\left(r\left(s_{t},a_{t}\right)+\gamma Q^{\prime }\left(s_{t+1},\mu^{\prime }\left(s_{t+1}|\theta^{\mu^{\prime }}\right)|\theta^{Q^{\prime }}\right)-Q\left(s_{t},a_{t}|\theta^{Q}\right)\right)}^{2},$$
where $r\left(s_{t},a_{t}\right)$ is the reward function, $Q^{\prime }\left(s,a\;|\theta^{Q^{\prime }}\right)$, and $\mu^{\prime }\left(s|\theta^{\mu^{\prime }}\right)$ is the target critic network and target actor network with parameters $\theta^{Q^{\prime }}$ and $\theta^{\mu^{\prime }}$. The overall minimization objective of DDPG is as follows:

In Algorithm 1, we use the prioritized experience to replay a statistical technique that enables the efficient utilization of experience to accelerate the convergence speed of the model. Furthermore, the algorithm leverages the addition of Gaussian noise to the decision-making process, enhancing the model’s exploratory power.
**Algorithm 1** DDPG-based trajectory planning.**Input:** batch size: B, discount factor γ, target smoothing coefficient τ, number of training episode: M, timesteps of each episode: T, exponents α and β of prioritization sampling, noise variance σ.**Initialize** actor network $\mu \left(s|\theta^{\mu }\right)$ and critic network $Q\left(s,a\;|\theta^{Q}\right)$ with parameters $\theta^{\mu }$ and $\theta^{Q}$.**Initialize** target actor network $\mu^{\prime }\left(s|\theta^{\mu^{\prime }}\right)$ and target critic network $Q^{\prime }\left(s,a\;|\theta^{Q^{\prime }}\right)$ with parameters $\theta^{\mu^{\prime }}$ and $\theta^{Q^{\prime }}$.**Initialize** prioritized replay memory B=⌀, Δ=0, $p_{1}=1$.forepisode=1:Mdo   Reset the environment and receive the initial observation state $s_{1}$.   fort=1:Tdo     Select action $a_{t}=\mu \left(s_{t}|\theta^{\mu }\right)+N\left(0,\sigma I\right)$ according to the actor network and exploration noise     Execute action $a_{t}$ and obtain the reward $R_{t}=r\left(s_{t},a_{t}\right)$ and the next observation state $s_{t+1}$     Store transition $\left(s_{t},a_{t},R_{t},s_{t+1}\right)$ in B with maximal priority $p_{t}=max_{i<t}p_{i}^{\alpha }$     Sample transition $j∼P\left(j\right)=p_{j}/\sum_{i}p_{i}$     Compute importance-sampling weight $\omega_{j}={\left(B\dot{P}\left(j\right)\right)}^{-\beta }/max_{i}\omega_{i}$     Compute TD-error $\delta_{j}=R_{j}+\gamma_{j}Q^{\prime }\left(s_{j},\mu^{\prime }\left(s_{j}|\theta^{\mu^{\prime }}\right)\right|\theta^{Q^{\prime })}-Q\left(s_{j-1},a_{j-1}|\theta^{Q}\right)$     Update critic by minimizing the loss (Equation 16)     Update actor using the sample policy gradient (Equation 15)     Update transition priority $p_{j}=\left|\delta_{j}\right|$     Accumulate weight-change $\Delta =\Delta +\omega_{j}{\dot{\delta }}_{j}\nabla_{\theta }Q\left(s_{j-1},a_{j-1}|\theta^{Q}\right)$     Update the target network:     $\theta^{Q^{\prime }}\leftarrow \tau \theta^{Q}+\left(1-\tau \right)\theta^{Q^{\prime }}$     $\theta^{\mu^{\prime }}\leftarrow \tau \theta^{\mu }+\left(1-\tau \right)\theta^{\mu^{\prime }}$   endforendfor

### 3.3. Optimization-Based Trajectory Planning

A trajectory from the starting point to the target point will be obtained quickly after trajectory planning via the DRL-based trajectory. Still, some trajectories penetrate obstacles due to the DRL-based trajectory, so it needs to be corrected via the optimization-based method. Because the reference point of the hot start is obtained via DRL-Based trajectory planning, optimization-based trajectory planning based on obstacle constraint can be solved quickly. Suppose T={x(0),x(1),x(2),…,x(N)} is the trajectory point obtained via soft constraint path planning. It is sampled before passing hard-constrained trajectory planning, which can be upsampling or downsampling, depending on the target accuracy of trajectory planning. Finally, the sampled trajectory point is obtained as $T_{sampled}^{ref}=\left\{x^{ref}\left(0\right),x^{ref}\left(1\right),x^{ref}\left(2\right),\ldots ,x^{ref}\left(M\right)\right\}$ where *M* is the number of trajectory points after sampling.

In optimization-based trajectory planning, this paper uses sampled trajectory points to constrain the reference curve of OBCA, which uses the exact vehicle geometry to develop collision avoidance constraints for tight maneuvers in a narrow space and the nonlinear optimization is formulated as follows:
(17a)$$\begin{array}{cc} \; & \min_{x^{\prime },u^{\prime }}L\left(X^{\prime },T_{sample}^{ref},U^{\prime }\right), \end{array}$$
(17b)$$\begin{array}{cc} s.t.\;\; & x^{\prime }\left(0\right)=x_{\mathrm{start}},u^{\prime }\left(0\right)=u_{\mathrm{start}}, \end{array}$$
(17c)$$\begin{array}{c} x^{\prime }\left(M\right)=x_{\mathrm{target}}, \end{array}$$
(17d)$$\begin{array}{cc} \; & x^{\prime }\left(k+1\right)=x^{\prime }\left(k\right)+f\left(x^{\prime }\left(k\right),u^{\prime }\left(k\right)\right)\cdot \Delta t, \end{array}$$
(17e)$$\begin{array}{cc} \; & x_{\min}\leq x^{\prime }\left(k\right)\leq x_{\max}, \end{array}$$
(17f)$$\begin{array}{cc} \; & u_{\min}\leq u^{\prime }\left(k\right)\leq u_{\max}, \end{array}$$
(17g)$$\begin{array}{cc} \; & -g^{⊤}\mu_{k}^{\left(l\right)}+{\left(A^{\left(l\right)}t\left(x^{\prime }\left(k\right)\right)-b^{\left(l\right)}\right)}^{⊤}\lambda_{k}^{\left(l\right)}>0, \end{array}$$
(17h)$$\begin{array}{cc} \; & G^{⊤}\mu_{k}^{\left(l\right)}+r{\left(x^{\prime }\left(k\right)\right)}^{⊤}A^{{\left(l\right)}^{⊤}}\lambda_{k}^{\left(l\right)}=0, \end{array}$$
(17i)$$\begin{array}{cc} \; & ∥A^{{\left(l\right)}^{⊤}}\lambda_{k}^{\left(l\right)}∥\leq 1,\lambda_{k}^{\left(l\right)}⩾0,\mu_{k}^{\left(l\right)}⩾0, \end{array}$$
(17j)$$\begin{array}{cc} \; & \mathrm{for}\;\mathrm{all}\;k\in \{1,\ldots ,M\}\mathrm{for}\;\mathrm{all}\;l\in \{1,\ldots ,L\}, \end{array}$$
where $X^{\prime }=\left\{x^{\prime }\left(0\right),x^{\prime }\left(1\right),\ldots ,x^{\prime }\left(M\right)\right\}$ and $U^{\prime }=\left\{u^{\prime }\left(0\right),u^{\prime }\left(1\right),\ldots ,u^{\prime }\left(M\right)\right\}$ are the trajectory and control sequence generated via hard constraint trajectory planning, respectively. $L\left(X^{\prime },T_{sample}^{ref},U^{\prime }\right)=Q\sum_{k=0}^{M}∥x^{\prime }\left(k\right)-x^{ref}\left(k\right){∥}^{2}+R\sum_{k=0}^{M-1}{∥u^{\prime }\left(k\right)∥}^{2}$ is the penalty term for the deviation distance from the reference trajectory and the size of the control magnitude. Equations (17g)–(17i) are the smooth and exact reformulations using Theorem 2 in [8].

### 3.4. Overall Scheme of Hierarchical Trajectory Planning with Deep Reinforcement Learning

The framework of Hierarchical Trajectory Planning with Deep Reinforcement Learning (HTP-DRL) is shown in Figure 4, which is mainly divided into two parts DRL-based trajectory planning and optimization-based trajectory planning. After obtaining the parking space information and the vehicle’s parameters, a reinforcement learning model is used to make decisions based on the observed state and a coarse trajectory from the start to the goal point is output. A vehicle dynamics model is used to update the state. Compared to traditional optimization-based online trajectory planning methods or sampling-based trajectory planning methods that require constant collision detection and the computational time overhead caused by non-linear solving, a rough trajectory from start to finish can be obtained quickly with the reinforcement learning model.

After obtaining the rough trajectory via DRL-based trajectory planning, it is necessary to use optimization-based trajectory planning to constrain it at some points to achieve collision-free automatic parking trajectory planning. Here, we refer to the method of OBCA, which can achieve more accurate collision avoidance by using exact vehicle geometry and since DRL-based trajectory planning has obtained the reference trajectory points, the computational speed of the method of OBCA can converge faster so that the results of automatic parking trajectory planning can be solved in a limited time.

## 4. Federated Learning-Based Model Training

The integration of communication technology and intelligent terminals has formed an intelligent transportation system aimed at improving the safety and efficiency of transportation. In order to realize an intelligent transportation system, the widespread application of artificial intelligence technology benefits from its ability to access the large amount of real-time activity data generated by traffic participants. Therefore, most AI-based intelligent transportation system solutions rely on centralized learning frameworks on data centers. Model training methods based on federated learning can have high privacy and low communication latency. Thus, federated learning plays an important role in handling privacy-protected distributed trajectory planning training based on reinforcement learning.

The proposed method for training a reinforcement learning model based on federated learning is designed to achieve a privacy-preserving centralized reinforcement learning framework, as shown in Figure 1b. Algorithm 2 presents the centralized reinforcement learning method under the federated learning framework in pseudo code form. At the start of the training process, *N* federated learning participant vehicles download global model parameters, including actor and critic network parameters, from the central server node via base stations. Subsequently, each participant trained their own model based on the data collected from their respective environments. Afterward, each intelligent agent uploads their own model parameters to the central server. The central server node then performs the model aggregate update for all models and distributes the updated models to all participants.
**Algorithm 2** The federated learning-based reinforcement learning model training.**Input:** number of federated learning participants N, parameter aggregate intervals L, number of training episodes M.**Initial** global actor network and global critic network parameters W(0).**Initial** participant actor network and participant critic network parameters $W_{n}\left(0\right),\left(n=1,2,\ldots ,N\right)$.forepisodei=1:Mdo   Each DRL participant downloads W(i−1) from the central server node.   forinterval=1:Ldo     Each DRL participant updates locally with $W_{n}\left(i\right)$ on the owner current observation   endfor   Each DRL participant uploads the trained model parameters $W_{n}\left(i\right)$ to the central server   Central server receives all weights $W_{n}\left(i\right)$ and performs the federated averaging for W(i)   Broadcast the averaged model parameters W(i)endfor

## 5. Experiment and Result

### 5.1. Environment Setup

In our experiments, we use some cases from the dataset https://tpcap.github.io/benchmarks/ (accessed on 14 April 2023) [27] for algorithm validation, as shown in Figure 5, where cases 1–3 are normal cases representing parallel, vertical and oblique parking cases, respectively, where no regular parking position is set and irregularly prevented obstacles are set to constrain. Cases 4 and 5 represent cases with extremely narrow parking spaces. In case 4, if using the sampling-based trajectory planning method, the accuracy required for sampling is particularly largely affected by the curse of dimensionality, which makes the sampling-based path planning method unable to search for a trajectory from the starting point to the target point in a limited time. If the optimization-based method is used directly, a high collocation point density needs to be set, which renders a large-scale mathematical programming problem. The kinematic parameters of the vehicle are listed in Table 1.

This paper mainly uses three deep reinforcement learning methods for validation, namely DDPG, SAC and PPO, and implements HALOES based on DDPG. The model parameters of the neural network are mainly shown in Table 2. In addition, ReLU and Adam are used as the activation function and the optimizer of the neural network. The experiments are trained and tested on a platform with RTX A4000 and Intel(R) Xeon(R) Gold 5320 CPU at 2.20 GHz, respectively. All experiments are conducted in Python on a Linux system.

### 5.2. Result and Discussion

The first part is used to verify the convergence performance of the reinforcement learning part of HALOES. The performances of DDPG, SAC and PPO in trajectory planning are compared. The test environment used is case 4, where 2500 episodes are trained in case 4 to compare the convergence of the final reward values, where the reward value is calculated by summing the reward values of all steps in each episode. As shown in Figure 6, better performance is achieved using DDPG than SAC and PPO because the use of the Actor-Critic network can effectively learn the optimal decision from the historical trajectory. To verify the effect of using relative position relations on reinforcement learning, we modified the relative positions in the observed states to absolute positions named DDPG-withoutRel and SAC-withoutRel. To be more specific, the input state of DDPG-withoutRel and SAC-withoutRel is changed as $S^{t}=\left\{P_{e}^{t},P_{g}^{t},v_{e}^{t},\delta_{e}^{t},V_{obs}^{t}\right\},$ where $P_{g}^{t}$ represents the absolute position of the goal point. Compared with DDPG-withoutRel and SAC-withoutRel, the use of the relative position relation, as implemented in the DDPG, has been shown to yield superior results compared to other approaches. By utilizing relative position information, the DDPG effectively reduces the dimensionality of the state space, independent of the agent’s absolute position. This reduction in state space dimensionality has been shown to enhance the generalization capabilities of the model, leading to improved performance across a range of scenarios. Compared to the offline training method of Actor-Critic networks such as DDPG and SAC, the performance of PPO using online training is poor and there is no significant improvement in training because the training method using online reinforcement learning does not record the historical optimal information, resulting in the model being explored locally, which makes the model unable to find the optimal gradient. This is due to the fact that online training does not allow the model to find the optimal gradient because the model keeps exploring locally. Because DDPG outperformed both SAC and PPO in terms of trajectory planning, subsequent experiments were carried out on the basis of DDPG.

The second part validates the convergence of federated reinforcement learning in HALOES compared to original reinforcement learning. The results presented in Figure 7 highlight the comparative performance of Federated Learning-based Deep Deterministic Policy Gradient (FedDDPG) with respect to the original DDPG algorithm in multiple training scenarios. Specifically, the DDPG algorithm was utilized for training a common model on randomly selected cases 1–5, while FedDDPG involved the collaborative training of a model across four participants in multiple environments, followed by the fusion of the trained models. The horizontal axis of the figure indicates the number of testing rounds, with the model being evaluated after every 20 rounds of training. The vertical axis represents the total reward achieved during each evaluation. The results demonstrate that FedDDPG achieves comparable or better performance than the original DDPG algorithm across all scenarios, with the former exhibiting a more stable and consistent learning curve over time. Due to the high level of privacy protection currently being pursued in intelligent transport systems, federated learning allows only model parameters to be shared without causing privacy issues for individual vehicles. It can be seen that the use of federated learning can achieve the same performance as the original DDPG while achieving data privacy protection. In (d,e) in Figure 7, we can see that the fluctuation of the DDPG trajectory planning based on federated learning is more stable after convergence compared with the original DDPG. The experimental results presented in Figure 7b–d demonstrate the susceptibility of federated learning models to perturbations in the early stages of training, particularly in the context of model fusion. Specifically, the observed instability in model performance during testing can be attributed to the sensitivity of the federated learning process to early fluctuations in model weights and updates, which can significantly impact the final model. However, as the number of training sessions increases, the federated learning model gradually becomes more stable, as evidenced by the improved performance observed over time. The above situation is more obvious in Figure 7d, where the model is updated by multiple participants making it too far from the original model, resulting in large performance fluctuations during the training phase, but after the model converges, stable performance can be maintained with less fluctuations compared to DDPG. This can be attributed to the ability of the federated learning process to aggregate increasingly larger and more diverse datasets over time, resulting in a more robust and accurate model that is better able to generalize to new scenarios. Figure 7a shows that FedDDPG has a more stable performance than the original DDPG, which has performance fluctuations after training.

The third part is used to verify whether the proposed HALOES can accomplish the trajectory planning task under narrow space. This paper mainly verifies five scenarios, cases 1–5. The scenarios of cases 1 and 4 are parallel parking spaces and case 4 is narrower compared to case 1. Cases 2 and 5 are vertical parking spaces and case 5 is narrower compared to case 2. Case 3 is an oblique parking scene. The first row of Figure 8 shows the output reference trajectory of the DRL method of the first layer in HALOES. The second row of Figure 8 shows the final planned trajectory of HALOES in cases 1–5. Figure 9 shows the vehicle kinetic parameters of HALOES and Opt+OBCA at any moment under cases 2–5; we mainly consider the following parameters: speed, acceleration, steering wheel angle and steering wheel angle rate. Based on the comparison of the results of the DRL method and HALOES, it can be seen that using the method of DRL can provide a reference trajectory to the target point for automated parking trajectory planning. Since there is no obstacle collision as a constraint in DRL, the trajectory will overlap with the obstacle, but it can be quickly corrected by the optimization-based method in the second layer of HALOES to generate a collision-free trajectory to the target point. In addition, based on the comparison of the kinetic curves of HALOES and Opt+OBCA, it can be seen that the trajectory planning with reference by reinforcement learning and the trajectory planning with an optimization-based reference can be kept within a reasonable kinetic constraint. Cases 2, 3 and 5 can all reach the target position through continuous control and the kinetic curve at the planning is smooth. The trajectory in case 4 needs to include multiple vehicle braking because it is necessary to make multiple round-trip operations when parking in a narrow space. According to case 4 in Figure 8 and the kinetic curves in Figure 9, we can see that the vehicle makes multiple round-trip operations. This is due to the fact that lateral parking in a narrow space requires continuous gear switching, which requires making full use of the vehicle’s steering angle to maintain a wide range. However, finally, the vehicle can eventually stop at the target position in compliance with the kinetic constraints.

To verify the advantages of HALOES in terms of planning time, we compared the average planning times of Hybrid A* as well as Opt+OBCA in cases 1–5. Where Hybrid A* planning results contain only trajectory points and without vehicle control parameters, additional speed planning is necessary to complete the task. As OBCA requires a warm start term to calculate the reference trajectory points, in this paper, we do not use the Hybrid A* approach to calculate the reference trajectory. Instead, we use an optimization-based approach to calculate the reference trajectory, where an artificial potential field is used to define the optimized loss function and the calculated reference trajectory is then processed via OBCA to obtain the trajectory points needed for vehicle execution. The trajectories planned via Hybrid A* and Opt+OBCA in cases 1–5 are shown in the second and third row of Figure 8, respectively, and the trajectory of case 4 could not be solved in a limited time due to the tendency of Hybrid A* to have dimensional collapse. As can be seen from Table 3, HALOES achieves significant speedups in cases 1–4. In case 5, the parking space is too narrow, resulting in the computation time of both HALOES and Opt+OBCA exceeding that of Hybrid A*, but HALOES uses a reinforcement learning model as a guide, thus making the computation faster than the Opt+OBCA approach. Since Hybrid A* only completes the trajectory point planning and then needs to perform velocity quadratic planning afterward, Hybrid A* still needs to complete the velocity quadratic planning in case 5, regardless of the short computation time. Therefore it can be considered that HALOES has a clear advantage in terms of planning time.

## 6. Conclusions

In this paper, to improve automated parking in narrow spaces, HALOES, a novel hierarchical trajectory planning method with deep reinforcement learning and optimization-based approach integration, is proposed to achieve computational accuracy and computational time trade-off. HALOES uses DDPG as a baseline and uses federated learning to train reinforcement learning models for privacy-preserving training of intelligent agent systems in intelligent transportation systems. HALOES proposes a novel hierarchical approach to trajectory planning, where the output of the reinforcement learning model is used for secondary optimization of the trajectory to achieve a trade-off between computational time and accuracy in trajectory planning. Extensive experiments have demonstrated that HALOES generalizes well to a wide range of scenarios and achieves the same accuracy as the reinforcement learning algorithm, and by verifying the planning time in a wide range of cases, it can be seen that HALOES is significantly better than Hybrid A* and OBCA.

The scenarios verified in this paper are all static obstacles, so the proposed method is limited when moving obstacles are present. Automatic parking trajectory planning under the influence of moving obstacles is a more realistic and complex problem. Therefore, in future research work, we will focus on the optimization method of automatic parking trajectory planning in the case of moving obstacles such as other vehicles and pedestrians, existing in narrow parking environments.

## Figures and Tables

**Figure 1 sensors-23-04087-f001:**
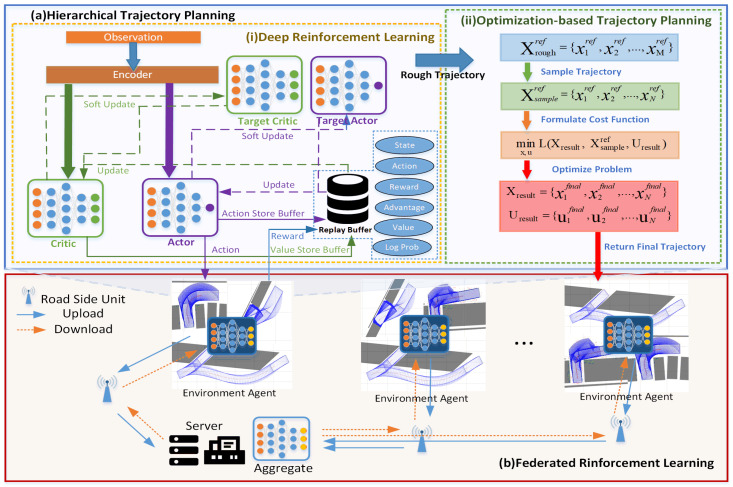
The scheme of HALOES contains the top and bottom parts of the figure, which are part (**a**) and part (**b**), respectively. Part (**a**) is the hierarchical trajectory planning, including the structure of deep reinforcement learning and the process of optimization-based trajectory planning. Part (**b**) is federated reinforcement learning, where different participating vehicles are involved in distributed reinforcement learning and trained deep reinforcement learning models in their respective environments. After that, the models from different vehicles will be aggregated at the central server.

**Figure 2 sensors-23-04087-f002:**
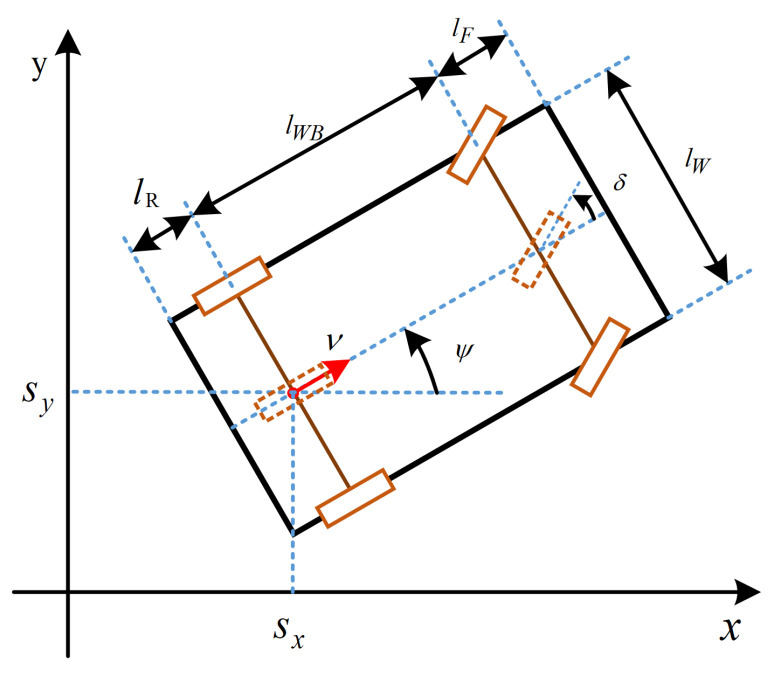
Kinematic Vehicle Model.

**Figure 3 sensors-23-04087-f003:**
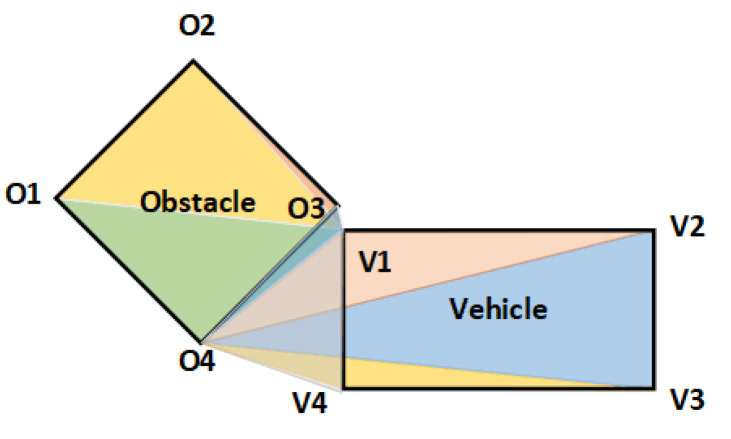
Diagram to determine whether a vehicle collides with an obstacle.

**Figure 4 sensors-23-04087-f004:**
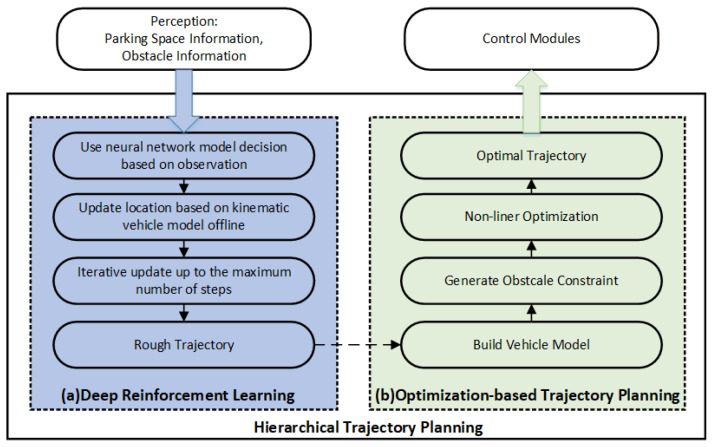
Overall structure of hierarchical trajectory planning with deep reinforcement learning.

**Figure 5 sensors-23-04087-f005:**
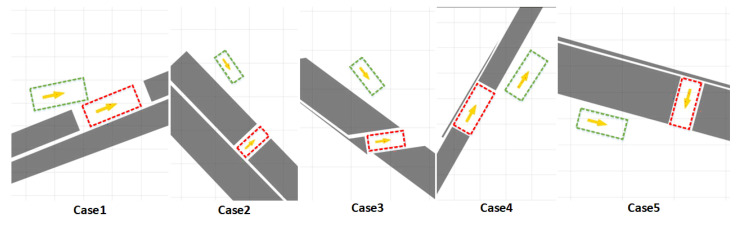
Experimental cases. The green box is the initial position of the vehicle and the red box is the target position of the vehicle.

**Figure 6 sensors-23-04087-f006:**
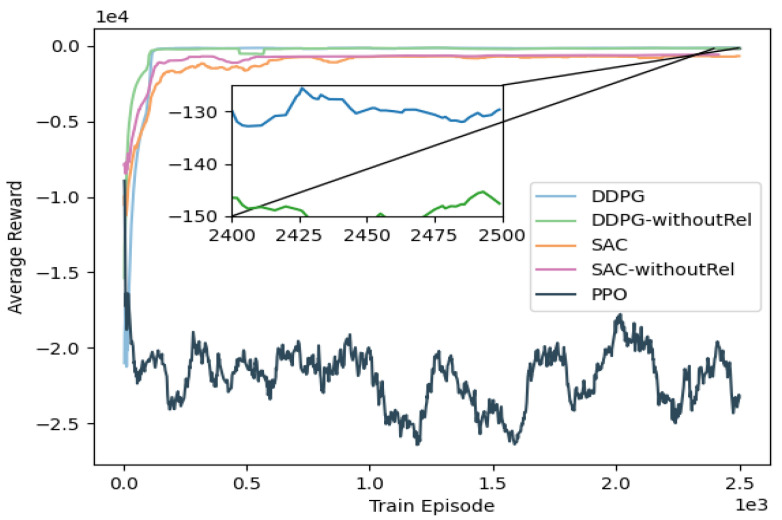
HALOES part I reinforcement learning training convergence, comparing DDPG, SAC and PPO under case 4.

**Figure 7 sensors-23-04087-f007:**
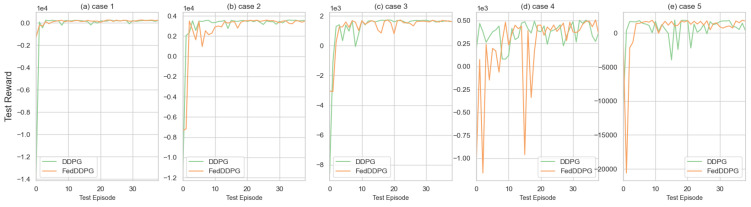
HALOES part I training convergence using federated reinforcement learning. Comparison of DDPG and Federated Learning-based DDPG under cases 1–5.

**Figure 8 sensors-23-04087-f008:**
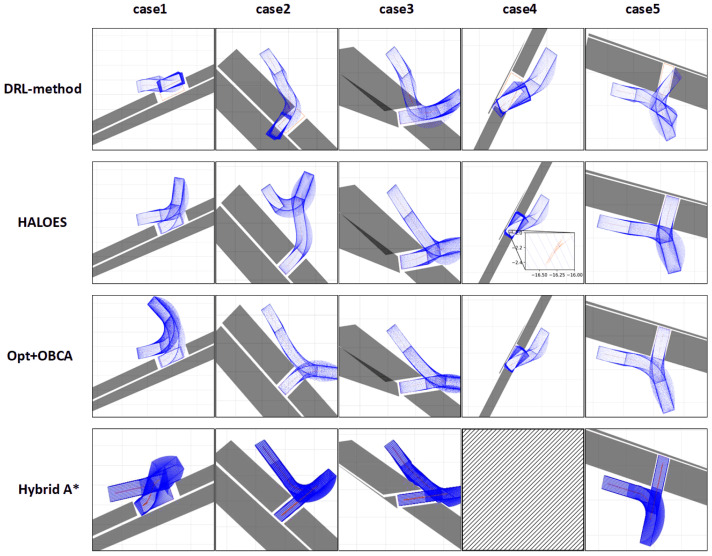
DRL method, HALOES, Opt+OBCA and Hybrid A* trajectory of the planning in cases 1–5.

**Figure 9 sensors-23-04087-f009:**
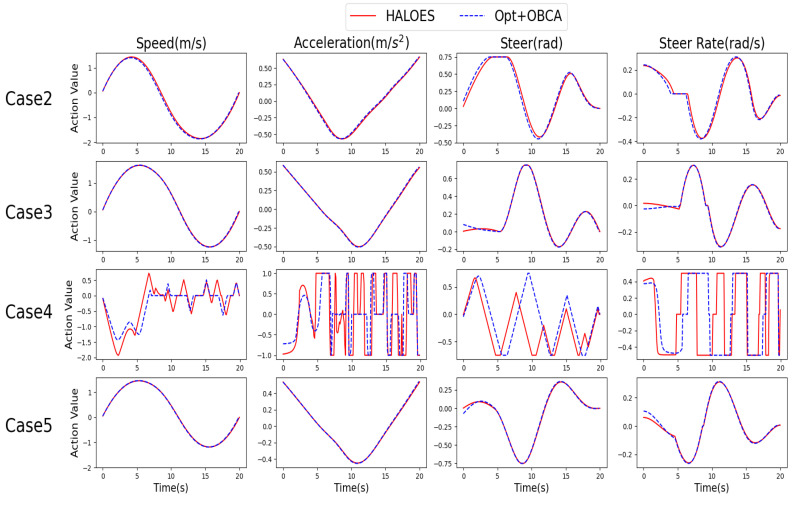
HALOES kinetic curve of the planned trajectory in cases 2–5.

**Table 1 sensors-23-04087-t001:** Parameters of Vehicle.

Parameter	Parameter Description	Value
$l_{wb}$	The wheelbase of vehicle	2.8 m
$l_{R}$	The rear hang length of vehicle	0.929 m
$l_{f}$	The front hang length of vehicle	0.96 m
$l_{w}$	The width of vehicle	1.942
$a_{max}$	The upper bound of acceleration	$1\;{m/s}^{2}$
$a_{min}$	The downer bound of acceleration	$-1\;{m/s}^{2}$
$v_{max}$	The upper bound of velocity	2.5m/s
$v_{min}$	The downer bound of velocity	−2.5m/s
$\phi_{max}$	The maximum steering angle	0.75 rad
$v_{\phi max}$	The maximum angular velocity of steering angle	0.5 rad/s
Δt	The control time interval of RL-based method	0.1 s

**Table 2 sensors-23-04087-t002:** Parameters of DDPG.

Parameter	Parameter Description	Value
*M*	The total episodes of training	2500
*T*	The total timesteps of each episode	800
*B*	The batch size of training	256
γ	The reward discounted factor	0.99
σ	The noise variance	0.01
τ	The target smoothing coefficient	0.05
$l_{actor}$	The learning rate of actor-network	$1\times 10^{-4}$
$vl_{critic}$	The learning rate of critic-network	$1\times 10^{-3}$
α	The parameter of prioritization sampling	0.6
β	The parameter of prioritization sampling	0.9

**Table 3 sensors-23-04087-t003:** Comparison of Hybrid A*, Opt+OBCA and HALOES in terms of planning time.

	Case 1	Case 2	Case 3	Case 4	Case 5
HybridA*	325.811 s	49.994 s	87.733 s	NULL	**26.067 s**
Opt+OBCA	42.805 s	42.805 s	40.932 s	75.723 s	35.546 s
HALOES(ours)	**27.955 s**	**14.546 s**	**17.159 s**	**36.359 s**	31.228 s
Improved over OBCA	34.69%	66.02%	58.08%	51.98%	12.15%

## Data Availability

The code is open-sourced at https://github.com/its-ant-bupt/HALOES (accessed on 13 March 2023).

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
