# Peer review of "Hierarchical Trajectory Planning for Narrow-Space Automated Parking with Deep Reinforcement Learning: A Federated Learning Scheme"

_sensors, 2023, doi:10.3390/s23084087_

Round 1
Reviewer 1 Report
This is a well-written manuscript on applying deep reinforcement learning and federated learning for narrow-space collision-free trajectory planning. The mathematical derivation, experimental design, and result plots are clearly illustrated. It will interest a reasonably broad readership of Sensors. I have the following comments that the authors should address before final acceptance:
1. Regarding previous publications and the state-of-the-art methods, except for the planning time benefit, what are the limitations of the proposed method in this manuscript? If the obstacles are pedestrian or motorcycles, or other types with more complicated shapes, how will the proposed method perform?
2. If the obstacles are moving, for example, other vehicles in this narrow space are moving, how would the proposed method perform comparing with state-of-the-art methods?
3. Figure 9 is hard to see and does not look scientific enough.
4. For case 4 in Figure 9, even the planning time looks short, the kinetic plots do not look convincing for a real world application. Please elaborate on this case.
Reviewer 2 Report
Dear Authors,
I am writing to provide some suggestions that could improve the quality and robustness of your work. Firstly, I have noticed a lack of references in your paper, and I would like to suggest some that you could include: "Data Efficient Reinforcement Learning for Integrated Lateral Planning and Control in Automated Parking System"; "Privacy-Aware Autonomous Valet Parking: Towards Experience Driven Approach"; "Federated Parking Flow Prediction Method Based on Blockchain and IPFS"; "Federated Learning in Vehicular Networks: Opportunities and Solutions"; "FedParking: A Federated Learning based Parking Space Estimation with Parked Vehicle assisted Edge Computing."
In the caption of Figure 1, I noticed that there is a mention of part A that is not visible in the figure. Therefore, it would be helpful to clarify the figure or provide more information in the caption.
I have also noticed that at Line 38, you assert that automated parking presents more challenges, but there are no references to support this claim. I would suggest including relevant sources to back up this statement. Similarly, in Line 44, it would be helpful to include references to support your claim.
At Line 55, I am curious why it is challenging to trade off computational accuracy with computational time. I would appreciate it if you could provide more information on this point.
Additionally, I have noticed that in Line 130, there is a mention of Deep Reinforcement Learning, and it would be helpful to include references to provide further explanation. Similarly, in Line 150, it would be useful to include references to support Federated Learning as a promising and innovative approach.
Regarding the S function (equation 6) mentioned in Line 206, I would like to suggest including another option for this function.
In Line 241, I noticed that there is no reference to the figure. Therefore, it would be helpful to include this information.
In section 5, it would be helpful to include more information about the hardware and software setup of the simulation. For example, which model was used, in which language was it coded, and on which PC and simulation platform was it tested?
In Figure 5, I have noticed that it is challenging to identify the green box because the border of the box is too small. Therefore, it would be helpful to increase the size of the border.
Finally, in Figure 9, it would be helpful to include the same graphs of the Opt+OBCA algorithm.
Reviewer 3 Report
See the attachment.

Round 2
Reviewer 1 Report
The authors have addressed all the points made in my first review report. I think it is acceptable for publication in the journal.